# Dry Fractionation and Gluten-Free Sourdough Bread Baking from Quinoa and Sorghum

**DOI:** 10.3390/foods12163125

**Published:** 2023-08-20

**Authors:** Regine Schoenlechner, Denisse Bender, Stefano D’Amico, Mathias Kinner, Sandor Tömösközi, Ram Yamsaengsung

**Affiliations:** 1Department of Food Science and Technology, University of Natural Resources and Life Sciences Vienna, Muthgasse 18, 1190 Vienna, Austria; regine.schoenlechner@boku.ac.at (R.S.); denisse.bender@boku.ac.at (D.B.); 2Austrian Agency for Health and Food Safety, Institute for Animal Nutrition and Feed, Spargelfeldstrasse 191, 1220 Vienna, Austria; stefano.d-amico@ages.at; 3Life Sciences and Facility Management, Zürich University of Applied Sciences, Einsiedlerstrasse 29b, 8820 Waedenswil, Switzerland; kinr@zhaw.ch; 4Research Group of Cereal Science and Food Quality, Department of Applied Biotechnology and Food Science, Budapest University of Technology and Economics, 1111 Budapest, Hungary; tomoskozi.sandor@vbk.bme.hu; 5Department of Chemical Engineering, Faculty of Engineering, Prince of Songkla University, Hat Yai 90110, Thailand

**Keywords:** dry fractionation, quinoa flour, sorghum flour, gluten-free bread, sourdough fermentation

## Abstract

The roller milling of sorghum and quinoa seeds into flour fractions (coarse, middle, and fine) was investigated, chemically analysed, and applied in the baking of gluten-free sourdough bread. The gap settings were adjusted to 0, 5, 8, and 10 for quinoa, and 3, 5, and 7 for sorghum. The fine fractions reached values of up to about 41% (gap 8) for quinoa and around 20% for sorghum (gap 5). SEM pictographs illustrated the clear separation of each fraction with the chemical analysis showing high contents of protein, TDF (total dietary fibre), and IDF (insoluble dietary fibre) in the coarse fraction. Up to 77% starch content was obtained in the fine fraction with significant amounts of SDF (soluble dietary fibre), which has good health benefits. Increasing the dough moisture up to 90% helped in decreasing the bread crumb firmness, while low Avrami parameters and RVA pasting behaviour indicated a slow bread-staling rate for both sourdough breads.

## 1. Introduction

Quinoa and sorghum have been gaining increased interest in the food industry, specifically in the baking industry. One reason for this growing interest is that both grains are very low in demand in terms of cultivation and can be grown in various environments, even in hot and dry areas, in cases where many cereals are less successful [1]. For this reason, they are generally acknowledged as food security plants of the future, particularly in view of the ongoing climate change. Another reason is that these two grains are gluten-free, and, thus, are suitable for the production of gluten-free products for people with intolerances to gluten-containing cereals like wheat, rye, and barley. This attribute has been one of the most important drivers for the rising cultivation and use of quinoa worldwide [2].

Bread and bakery products are one of the most common staple foods worldwide. Most of these bakery products are based on wheat, but due to the ongoing climate change and its already noticeable effects on wheat cultivation (yield and quality of harvested grains), an increase in biodiversity is needed. Hence, an integration of a wider range of grains might be necessary to maintain the supply of these products in the future. With respect to quinoa and sorghum, they both lack network-forming proteins and are, therefore, not able to develop cohesive and elastic dough properties when used alone [3,4,5]. Their use in baking is either carried out by blending them with wheat, or by adopting gluten-free recipes. Gluten-free breads in the market are often produced from refined flours (like rice) or pure starches (like maize, wheat, or potato starch), and they still often lack protein and dietary fibre. The use of wholemeal flours from gluten-free sources like sorghum or quinoa would be a beneficial option, as they provide valuable nutrients [6,7].

There is a bulk of research studies available on the use of quinoa in (gluten-free) baking [8], but, in the market, such bakery products containing quinoa are still niche products. The reasons for this might be that the addition of quinoa to bakery products is noticeable, altering the final product quality, particularly in terms of the texture and taste [8]. Additionally, the higher price of quinoa makes it more difficult for its integration into daily staple foods. Sorghum might be another interesting grain that could offer the potential to provide affordable and palatable bakery products. Sorghum has been a major staple food in Africa and India, where the local food processing of sorghum often includes microbial fermentation with the aim of improving the nutritional and sensory properties of the final products. Outside of Africa and India, sorghum is mainly used as animal feed or bioethanol production, and only rarely as food [9]. Hence, the use of sorghum in Western-style bakery products is still under-researched. The yield of sorghum is comparable or even higher than that of wheat, which might allow us to commercialise it at similar costs.

Up until now, both sorghum and quinoa are mainly used as wholemeal flours. As their grain size, depending on their genotypes, and also their kernel morphology, differs greatly from cereals like wheat, the production of milling fractions at a large scale has not yet been established. However, the production and use of chemically defined and distinct fractions might enhance the future use of these two kernels for a wider range of products. Compared to wholemeal flour, an increased level of starch or protein or chemical components might be of help. For the milling and fractionation of these two grains, the protocols from cereal milling require specific adaptations, on the one hand, and, on the other hand, a thorough knowledge of the composition of such produced flour fractions is necessary in order to select the desired fraction with a targeted composition or properties [10].

In order to support the future exploitation of sorghum and quinoa, this study aimed to investigate if milling fractionation and sourdough technology might be feasible tools for enhancing their potential for a broader food use. In detail, the research tasks of this study were (1) to investigate the production of chemically distinct milling fractions by roller milling, and (2) to explore the baking properties and shelf-life of gluten-free bread produced from 100% quinoa or 100% sorghum by sourdough fermentation. The use of the same methods for all trials allowed us to compare the behaviour and performance of sorghum and quinoa for the investigated processes.

## 2. Materials and Methods

### 2.1. Raw Material

Quinoa was obtained from Naturmühle Caj. Strobl (Linz-Ebelsberg, Austria) with the grains originally coming from Bolivia. Three sorghum species (Albita, Alfödi, and GK Emese) grown in Hungary were provided by the Research Group of Cereal Science and Food Quality, Department of Applied Biotechnology and Food Science, Budapest University of Technology and Economics, Hungary. All seeds were stored in sealed plastic bags at 4 °C.

All chemical reagents were either purchased by Merck KGaA, Darmstadt, Germany or Carl Roth GmbH + Co. KG, Karlsruhe, Germany.

### 2.2. Chemical Analyses

Chemical analyses were performed using ICC standard methods. Dry matter was determined according to ICC Standard 110/1 [11], the ash content according to ICC Standard 104/1 [12], starch according to ICC Standard 168 [13], protein according to ICC Standard 105/2 [14], and dietary fibre according to ICC Standard 156 [15]. All chemical analyses were performed in triplicate.

### 2.3. Pasting Properties

Pasting properties were evaluated according to the ICC standard method No. 162 [16] using a rapid viscoanalyser (RVA) 4500 (PerkinElmer Inc., Waltham, MA, USA). The samples were prepared by dispersing 3.5 g of flour (14% (*w*/*w*)) in 25 mL of distilled water. Determinations were replicated at least three times and the results were shown as mean values.

### 2.4. Dry Fractionation by Roller Milling

For fractionation of the samples, a laboratory scale roller mill (E8, Haubelt Laborgeräte GmbH, Istanbul, Turkey) was used. The mill consists of a pre-crash system where the seed kernels are broken, and two plain drums where the gap (distance of drums) is variable (0 for the widest possible gap and 10 for the closest gap, which is less than 1 mm). After milling, the flour was sieved through two vibrating sieves, which separated the flour into three different milling fractions: (1) coarse fraction, (2) middle fraction, and (3) fine fraction.

In pre-trials, several gap settings were investigated with the aim to obtain distinct milling fractions, which were evaluated by determination of the ash content. Also, yield of the fractions was considered. For quinoa, fractionation gap settings investigated were 0, 5, 8, and 10; while, for sorghum (only variety Albita), the gap settings were 3, 5, and 7. The setting of the pre-crasher was kept constant at 6, as well as the time for sieving at 2 min. Mesh size of the sieves used were 475 µm (top) and 212 µm (bottom). Pre-trials were run once, but ash content of all obtained fractions was measured in triplicate. According to the results of the pre-trials, the main trails were run at a reduced number of gaps, which was 5 and 8 for quinoa, and 5 for the three sorghum varieties. All main milling trials were performed in triplicate. No conditioning of the kernels was performed prior to milling.

### 2.5. Gluten-Free Bread Baking

Gluten-free bread baking was performed using the sourdough technology as suggested by Ramos et al. (2021) [17]. Additionally, water addition was varied at 70–80–90%, as gluten-free batters are known to require a higher level of water. In the first step, the sourdough was pre-fermented for 16 h. For the preparation of sourdough, flour (quinoa or sorghum flour) and water was mixed in a ratio of 1:1 with 10% sourdough starter (Reinzucht-Sauerteig Reis glutenfrei, Ernst Böcker GmbH & Co. K.G., Minden, Germany), which was then fermented at 27 °C, 80% RH for 16 h. The sourdough was immediately used for bread baking. Bread dough was prepared by mixing 100% sourdough (flour to water ratio = 1:1) with 75% flour (mixture of quinoa or sorghum with gluten-free wheat starch in a ratio of 1:1), 3% albumin (Enthoven-Bouwhuis Eiprodukten B.V., Raalte, The Netherlands), 1.5% fat powder (REVEL^®^, Loders Croklaan B.V., Wormerveer, The Netherlands), 1.8% salt, 1.5% instant dry yeast (Lesaffre, Marq-en-Bareoul, France), and 1% sugar (percentage was based on total flour weight). Water was added to produce a batter with an overall dough moisture of 70, 80, and 90% moisture content (w.b.), considering the water amount of the sourdough. The batter was mixed in a laboratory dough mixer (Teddy Varimixer, Varimixer A/S, Brondby, Denmark) at speed 1 for 2 min, followed by speed 4 for 4 min, then divided into two portions of 300 g, which were placed in baking tins with dimensions of 13 × 9 × 7 cm (top dimension) and 15 × 11 × 7 cm (L × W × H; bottom dimension). Proofing was performed in a fermentation chamber (Model 60/rW, MANZ Backtechnik GmbH, Creglingen, Germany) at 30 °C and 85% RH (relative humidity) for 50 min. Baking was conducted in a deck oven (Model 60/rW, MANZ Backtechnik GmbH, Creglingen, Germany) at 180 °C (top and bottom heat) for 40 min. After baking, the bread was cooled and stored in a climate chamber (Climacell^®^ EVO, MMM Gmbh, München, Germany) at 20 °C and 50% RH for 18 h before analysis. Baking was conducted in triplicate, resulting in six bread loaves per recipe.

### 2.6. Storage Tests of Gluten-Free Bread

Five bread loaves of one quinoa and one sorghum recipe, both at 90% batter moisture, were baked as described above and stored at 20 °C, 50% RH. After 1, 3, 5, 7, and 9 days, the crumb firmness was determined. The obtained Fmax values were used to describe bread staling (caused by starch retrogradation) using the equation of Avrami (Equation (1)).
(1)$$Ɵ=\frac{T_{inf}-T_{t}}{T_{inf}-T_{0}}=e^{-ktn}$$
where Ɵ = starch ratio, which has not re-crystallised; *T*_0_ = initial F_max_ at day 0; *T_inf_* = final F_max_ at day 9; *T_t_* = F_max_ at time “*t*“; *k* = rate constant; and *n* = Avrami exponent.

### 2.7. Evaluation of Physical Bread Properties

The baking loss was calculated as (Wbb − Wab)/Wbb × 100, where Wbb refers to the mass of the batter and Wab refers to the mass of the bread after baking and cooling.

Measurement of bread volume was performed by applying the BVM 6600 volume analyser (PerkinElmer Inc., Waltham, MA, USA). Specific bread volume was calculated as the ratio between volume (cm^3^) and the bread (g) mass. Four measurements were performed for each formulation.

Determination of crumb firmness and relative elasticity was conducted based on the AACC Method 74–09.01 with some adaptations: a Texture Analyser (Model TA-XT+ Stable Micro systems^TM^ Co., Godalming, UK) equipped with a 5 kg load cell and a SMS compression probe of 100 mm diameter (SMS P/100) was used to analyse two bread slices of 3 cm thickness per loaf for crumb texture with a uniaxial compression test of 20% strain. The test speed was set at 0.5 mm s^−1^ with a relaxation time of 120 s and trigger force of 10 g. Pre- and post-test speeds were set at 1 and 10 mm s^−1^, respectively. The maximum compression force (N) was indicated as crumb firmness F_max_, and relative elasticity (%) was determined by dividing the end force by the maximum force. Six values were derived for each recipe.

Crumb and crust colour were measured by applying a Digi-Eye^®^ system (Verivide, Leicester, UK) with a D-90 Nikon digital camera (Tokyo, Japan), resulting in L*, a*, and b* values. Four values for each formulation were determined.

A digital image analysis system was applied to measure crumb porosity by using the software ImageJ (1.47v, National Institute of Health, Bethesda, MD, USA). The analysis was performed on an image of a 2 × 2 cm crumb square, captured with the digital camera features of D-90 Nikon (Tokyo, Japan) from the Digi-Eye^®^ System (Verivide, Leicester, UK). The analysed parameters included number of pores (*n* = 4), average pore size (*n* = 8), percentage of total pore area to total bread area (*n* = 8), and pore uniformity (calculated from the standard deviations of the average pore size per bread, *n* = 4).

### 2.8. Scanning Electron Micrographs

To determine the particle details after fractionation, a scanning electron microscope FEI Inspect S50 (FEI Company Japan Ltd., Tokyo, Japan) at 50× magnification was used.

### 2.9. Statistical Analysis

Statistical analyses were conducted using the software Statgraphics Version XIX (StatPoint Technologies, Inc., Warrenton, VA, USA). To express significant differences between formulations, a one-way ANOVA and Fisher’s least significant difference test at a 5% probability level (*p*-value < 0.05) were applied.

## 3. Results and Discussion

### 3.1. Milling Performance—Yield of Obtained Fractions

As described in the Section 2, a series of pre-trials was conducted where several roller gaps were investigated for their applicability in producing distinct milling fractions. For a rough estimation of successful separation, the ash content was determined for the obtained fractions (see Table 1). The ash content is an appropriate predictor with which to evaluate the separation of the outer layers (bran, aleurone, and embryo) from the inner kernel (starch endosperm) [18]. In Figure 1, the yield of fractions gained at selected milling gaps is summarized.

For all samples, a too-wide and too-closed gap between the rollers was not suitable. At gap 0 (widest gap of the rollers), the quinoa fractions were not clearly separated, and the amount of coarse fraction was too high and, therefore, containing large amounts of endosperm, which was also shown by the rather low ash content. Decreasing the distance of the rollers improved the fractionation process, but, when it was too small (gap 10), the separation of the samples worsened again. The ash content in the coarse fraction was decreased compared to setting 5 and 8. For quinoa, these pre-trials suggested that gap 5 or 8 seemed most suitable. As the differences between these two gaps were low and, at this stage, not completely clear, we decided to run the main trials for quinoa at gap 5 and 8. For sorghum fractionation, the results were clearer compared to quinoa: only gap 5 delivered fractions with a distinctly different ash content. At gap 7 (closer), the ash content of the fine fraction was higher and the ash content of the coarse fraction was lower than at gap 5. However, at gap 3 (wider), too much of the coarse fraction was accumulated, containing obviously too much endosperm. Thus, the main trials for the sorghum varieties were run at gap 5.

The fraction yields of the main trials can be derived from Figure 2. Scanning electron microscopy (SEM) pictures of these fractions are shown in Figure 3. The results demonstrate that all milling trials were very reproducible, and the standard deviations for all samples were rather low; also, the results between the two set of trials (pre- and main trials) were comparable.

For quinoa, the separation into distinct milling fractions was not as differentiated as is usually achieved for cereals like wheat. The fine fraction reached only values of up to about 41% (gap 8), and, at both gaps, high amounts of the coarse (bran) fraction were gained (up to almost 23% with gap 5). The middle fraction was obtained in similar amounts to the fine fraction (gap 8), with gap 5 yielding an even higher amount than for the fine fraction. This different fraction performance of quinoa compared to other cereals (wheat) is most likely a result of the different botanical morphology of the seed. In quinoa, a relatively large embryo surrounds the kernel in the form of a ring and makes up about 25% of the seed weight [19]. Most of this embryo was found in the bran fraction (as desired), but some broken fragments of it passed through the first sieve to the middle fraction as was visible in the SEM photos (Figure 3). The low amount of the fine fraction arises from the fact that the seed’s proportion of the starchy perisperm is smaller compared to the proportion of the endosperm in cereals.

In the sorghum roller milling trials, high amounts of coarse fractions were found as well (around 23% for all varieties), but, in contrast to quinoa, even lower amounts of the fine fractions were obtained, as the middle fractions comprised the highest amounts (54–59%). Although sorghum is botanically a true cereal, with its small kernel size and rather large embryo (though within the kernel, in contrast to quinoa), it seems rather challenging to separate the milling fractions and to obtain sufficient amounts of the fine fractions. For industrial milling, additional milling passages of the middle fraction would be necessary to increase the amount of fine endosperm flour.

### 3.2. Chemical Composition of the Milling Fractions

All obtained roller milling fractions were analysed for their content of protein, starch, ash, and dietary fibre (total, insoluble, and soluble fibre) in order to evaluate whether the milling fractions can be classified into chemically distinct flours. The results are presented in Figure 4 and Figure 5.

As already described above and also by other researchers [4], quinoa fractionation is challenging. Still, the analytical results for the milling fractions obtained with the two different milling gaps demonstrated that they chemically differ from each other. The coarse fraction contained the highest amounts of protein, ash, and dietary fibre, while starch was abundant in the fine fraction. The middle fraction showed more or less values in between these two fractions, which was more obvious for fractions milled at gap 5 than at gap 8. All chemical components, starch, protein, dietary fibre, and ash, were found in similar amounts in the middle fraction as in the whole seed. The fine fraction had the lowest amounts of ash and dietary fibre.

Compared to wheat milling, the quinoa fractions were still not clearly separated, but they can still be considered to be distinct fractions. When comparing the coarse and the fine fraction, the coarse fraction in quinoa contained 4–4.6 times more protein, 5.7–8 times more ash, 8.6–9 times more TDF, and only half the amount of starch. The middle fraction, which was chemically not really different from wholemeal flour, still contains too many outer layers and embryo particles. Additional milling and sieving/sifting passages would allow a further extraction of the starch endosperm from the kernel.

Looking at the results of the sorghum fractions, it has to be stated that, through the application of this roller milling system, it was even more challenging to produce distinct milling fractions than it was for quinoa, but still some interesting outcomes were achieved. As the data for yield have shown, only small amounts of fine fractions were obtained, which hardly differ from the middle fraction or wholemeal flour in terms of the ash content, although ash was accumulated, to some extent, in the coarse fraction, where the amount was about twice as much compared to the other flour fractions. Starch was the chemical component which was least separated into the collected sorghum milling fractions. Obviously, too much of the inner (endosperm) part of the kernel remained in the coarse fraction (starch still made around 50 g/100 g dm), which is also reflected by the high yield of this fraction. The amount of protein was descending from the coarse fraction to the fine fraction. Interesting results were found for dietary fibre: the majority of the TDF was found in the coarse fraction, and both the middle and fine fractions contained rather low amounts. But, in contrast to quinoa, it was mainly the insoluble dietary fibre fraction (IDF) that was concentrated in the coarse fraction, while a large proportion of the soluble dietary fibre (SDF) was enriched in the fine fraction, significantly higher than in the middle fraction. As soluble fibres are known to have beneficial health effects, sorghum endosperm flour with such high SDF amounts might have a future potential for food applications. Yet, in order to improve the yield of the fine fraction and improve the separation of starch (endosperm parts) from the coarse fractions, sorghum milling requires further milling and sieving/sifting passages. Kebakile et al. (2007) assessed different sorghum milling technologies for product development [20]. Also, Mezgebe et al. (2020) coupled roller milling with hammer milling for making fermented flat bread of sorghum genotypes [21]. Furthermore, such roller milling trials with several passages have been performed by Rumler et al. (2021) with sorghum on a pilot scale [10]. Similarly, they obtained fractions with increased bran material that contained higher amounts of ash, protein, fat, total dietary fibre, and total phenolic content, but less starch. The additional determined physical quality parameters revealed that these fractions with higher amounts of bran showed enhanced water absorption and water solubility indices. But, also, in their study, the fine and middle fractions were not always chemically distinct. For further industrial milling, conditioning of the kernels (i.e., moistening of the kernels overnight) prior to milling might improve the separation process of the bran from the kernel.

### 3.3. Pasting Properties of the Milling Fractions (RVA)

Figure 6 shows the viscosity measurements of all fractions compared to wholemeal flour. All curves are average values of triplicate determinations.

In general, all quinoa fractions showed lower viscosity values than the sorghum fractions. Within the quinoa fractions, the finest fraction showed the highest viscosity values, while the curves for the middle and coarse fraction were almost identical and lower than the wholemeal flour. In quinoa, no breakdown in viscosity (decrease during the holding period) was observed, which indicates their high hot paste stability. The final viscosity (viscosity at the end of the cooling period) decreased for all fractions; they were all showing low setback.

The pasting properties help to indicate baking quality, since there is a tight relationship between the rheological and mixing properties of flour/water mixtures, e.g., dough strength/torque, water absorption, shear and temperature stability, crumb firming, and storage behaviour [22,23]. Peak viscosity is associated with the final product quality; high peak viscosity during pasting and low viscosities after the holding period at 95 °C are considered predictors of bread-firming behaviour during storage, and low setback viscosities indicate low rates of starch retrogradation in bread baking [23]. Looking at the pasting properties of quinoa, they offer a good potential for improving the shelf-life of baked products.

Within the sorghum fractions, only the fine fractions were distinct from the other fractions (higher). Wholemeal flour and the middle fraction were more or less very similar and not clearly different from each other, and coarse flour fractions showed the lowest viscosities, except for Albita; here, wholemeal and coarse flour fraction were alike. Between the three species, no relevant differences were observed: the viscosity of the Albita fractions was always slightly lower than that of the two other species. In contrast to quinoa, all sorghum species showed a small breakdown of the hot paste viscosity, especially the fine fractions, and all flours developed an increase in final viscosity.

Considering the chemical composition of the flour fractions, the starch content alone was not really a determinant for RVA viscosity in quinoa. From this point of view, one would expect rather similar curves for wholemeal flour and the middle fraction, which often showed a similar composition. In quinoa, the fine fraction contained the highest amount of starch, the least amount of protein, TDF, SDF, and IDF; thus, its higher viscosity is expected. But, looking at the middle and coarse fraction, which showed similar RVA curves, the chemical composition gives no clear explanation as these two fractions were not chemically similar. A suggestion could be that, although the starch content in the coarse fraction is lower compared to the middle fraction, the coarse fraction contains a much higher amount of TDF, which obviously contributes to its viscosity development, in particular, for SDF. In sorghum fractions, the starch content in the fine fraction was rather similar to wholemeal flour or to the middle fraction (only higher compared to the coarse fraction), but they showed significantly higher viscosity. The fine fraction contained less protein, less TDF, less IDF, but more SDF compared to the middle fraction or wholemeal flour, so it is the sum of all these differences which determined the RVA pasting viscosity and, thus, its food uses.

### 3.4. Bread-Baking Quality

Gluten-free baking trials were performed from either 100% quinoa or 100% sorghum, both of them by sourdough fermentation. Sorghum and quinoa food use is improved by microbial fermentation, and previous research suggests that this technology has advantages for gluten-free bread baking in general [17]. The results of the physical bread quality obtained from these baking trials are summarized in Table 2, results for the shelf-life are shown in Table 3 and Figure 7, and bread slices are visualized in Figure 8.

As could be observed in this study, increasing dough moisture from 70 to 90% resulted in gluten-free breads with improved bread quality parameters, which was even more pronounced for sorghum breads than for quinoa breads. Baking loss showed significant differences between the different levels of water addition. In both breads, from quinoa and sorghum, a higher water addition resulted in higher baking losses. Thus, after baking, all breads contain about similar amounts of dough moisture, as a higher water addition to the dough was balanced out by higher baking losses. Specific volume was higher in the sorghum breads than the quinoa breads and increased with a higher water addition in both of them. Despite the difference in specific volume, crumb firmness was similar between the quinoa and sorghum breads. A higher water addition decreased firmness, which, in the quinoa breads, also increased relative elasticity. Similar results were found by Hera et al. (2014) [24] for gluten-free rice breads; also, in their study, baking loss and specific volume increased and crumb firmness decreased when dough moisture was increased from 70% to even 110%. The determined pore properties reflect the difference between the quinoa and sorghum breads. The total pore area was higher in sorghum than in quinoa bread (see Table 2). Quinoa breads showed a rather dense crumb structure, and were characterized by a high number of uniform pores of smaller average size, while sorghum breads had a much lower number of large average-sized pores, but the pore size was very irregular. Dough moisture hardly influenced the pore average size and the number of pores, but pore uniformity was more irregular at higher moisture, which was also seen in gluten-free breads from amaranth [24].

Bread crumb colour is mainly influenced by the intrinsic colour of the ingredients, while the crust is caused by the extent of the Maillard reaction during baking. Higher baking temperatures and longer baking times accelerate this reaction, while higher dough moisture reduces this browning reaction. This was also observed in this study: higher moisture produced a slightly paler crust. Quinoa breads were generally much lighter, but had higher yellow-colour values in the crust and crumb, and higher red values in the crust but lower in the crumb compared to the sorghum breads. Figure 8 illustrates that sorghum breads were much darker than quinoa breads. This might influence consumer acceptance, but, in countries where higher rye bread consumption is more common, this darker colour could well be an advantage.

One parameter that often diminishes the quality of (commercially available) gluten-free bread compared to gluten-containing ones is that they tend to stale much faster, and, thus, have a reduced shelf-life. Among others, this is a result of the higher starch and lower protein and dietary fibre content in gluten-free breads, which are often baked from pure starches or polished rice flour. Quinoa and sorghum were used as wholemeal flours in this study; additionally, sourdough fermentation was applied, which is known to prolong crumb softness in rye or wheat breads. In order to determine their shelf-life, storage tests of quinoa and sorghum breads produced with 90% dough moisture were undertaken. The obtained crumb firmness values are presented in Figure 7, and the results of the Avrami parameters are summarized in Table 3. From the results, it can be seen clearly that the staling rate of both breads was slow, also demonstrated by their low Avrami exponent (n) or rate constant (k) value. High values of n or k indicate a faster staling rate, although high n and k values do usually not occur combined [25]. Sorghum breads showed a higher initial crumb firmness than quinoa breads, but both breads doubled this value only after 7 days. The previously described pasting properties (low setback and peak viscosity) already suggested that both grains, in particular, quinoa, possess slow starch retrogradation rates, and, thus, can retard bread firming during storage.

## 4. Conclusions

The study showed that sorghum and quinoa flour and milling fractions have interesting properties for food application. The trials performed on a lab scale provided some preliminary insights into the production of flours with targeted properties, e.g., enrichment of certain nutrients, or adaptation of physical properties. Starch-rich fine fractions and protein/TDF-enriched coarse fractions were obtained, and, interestingly, in the sorghum fine fraction, SDF was abundant. For future upscaling, further research is needed. The determination of the pasting properties revealed that both grains, quinoa more than sorghum, possess slow starch retrogradation tendencies, which is an advantageous feature for bakery products.

With respect to baking standards, sorghum breads with an acceptable volume and crumb structure were obtained. In quinoa bread, the crumb was slightly denser; the addition of higher amounts of protein or emulsifiers might enhance quinoa bread texture. Both grains, and probably also the application of sourdough fermentation, produced breads with a prolonged shelf-life of up to one week, which is a clear benefit for consumers.

## Figures and Tables

**Figure 1 foods-12-03125-f001:**
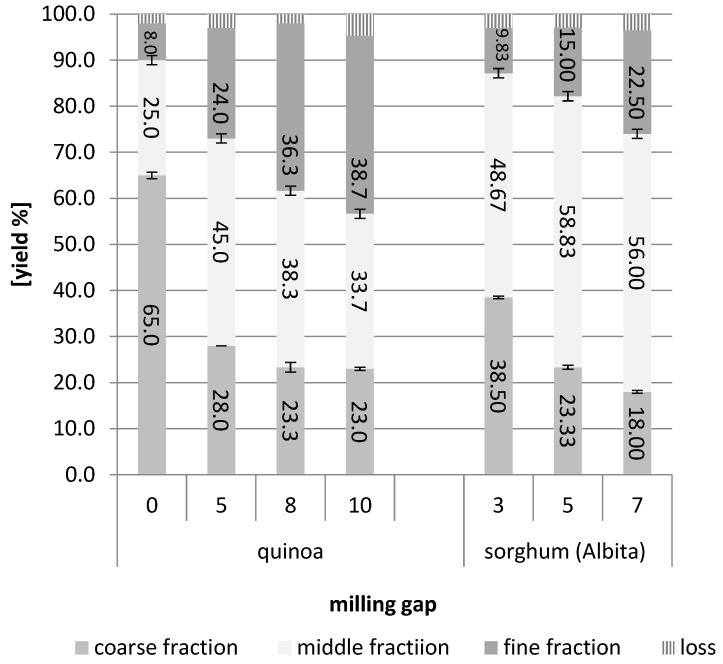
Milling pre-trials for quinoa and sorghum at different milling gaps (=distance of rollers): yields (%) obtained for fine (<212 µm), middle (212–475 µm), and coarse (>475 µm) fractions.

**Figure 2 foods-12-03125-f002:**
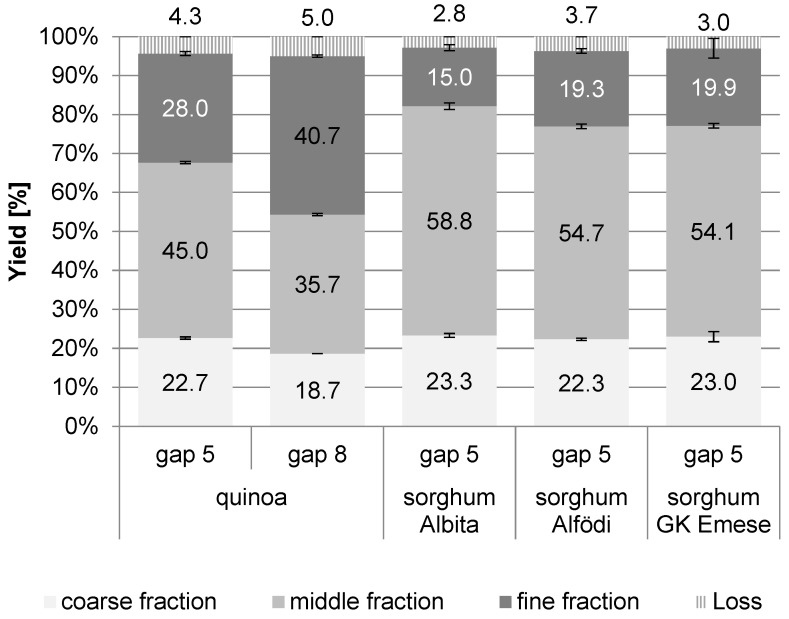
Milling main trials for quinoa and sorghum: yields (%) obtained for fine (<212 µm), middle (212–475 µm), and coarse (>475 µm) fractions. Quinoa was milled at two different milling gaps (5 and 8); sorghum at gap 5.

**Figure 3 foods-12-03125-f003:**
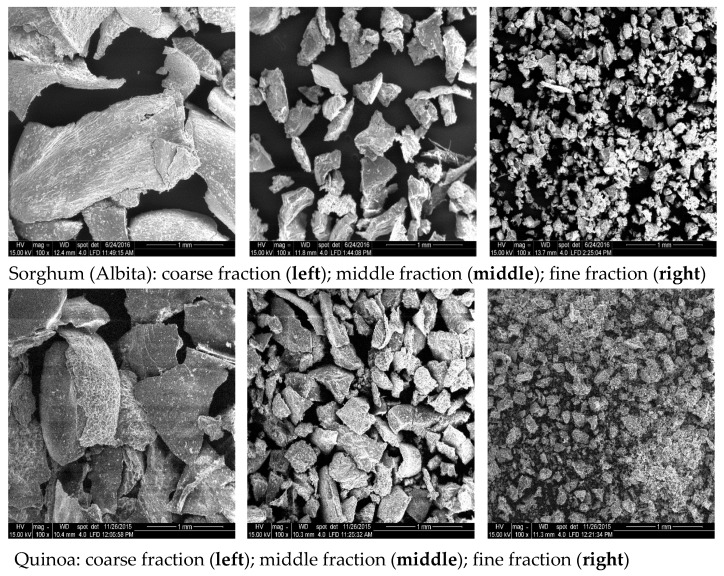
Scanning electron microscopy pictures of the obtained milling fractions, gap 5 (sorghum top, quinoa bottom).

**Figure 4 foods-12-03125-f004:**
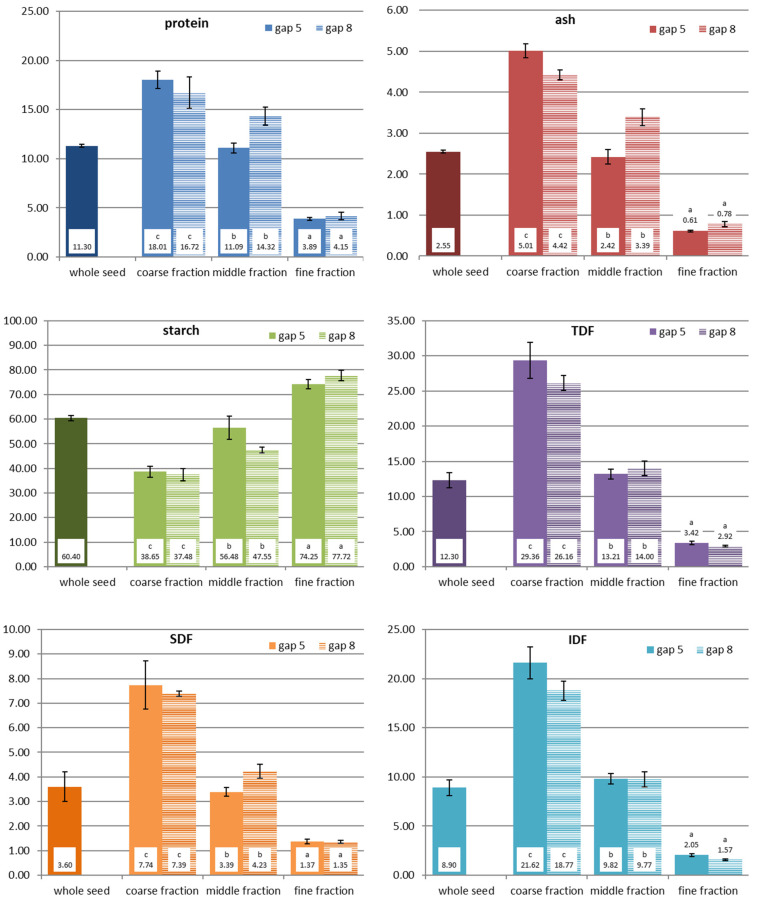
Chemical composition of quinoa fractions milled at two different gap settings (gap 5—wider gap; gap 8—closer gap); results are presented as average values and standard deviation of triplicate measurements (% dm). Different lowercase letters before the values indicate significant differences between the milling fractions (within same gap setting).

**Figure 5 foods-12-03125-f005:**
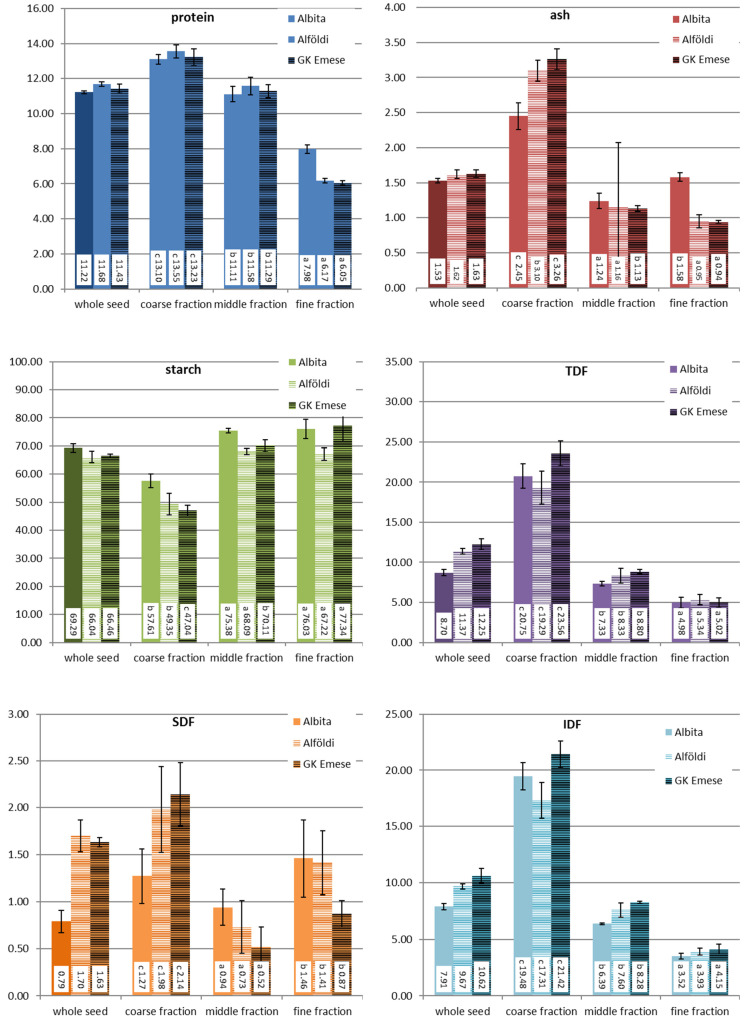
Chemical composition of sorghum fractions milled at gap 5; results are presented as average values and standard deviation of triplicate measurements (% dm). Different lowercase letters before the values indicate significant differences between the milling fractions (within same sorghum species).

**Figure 6 foods-12-03125-f006:**
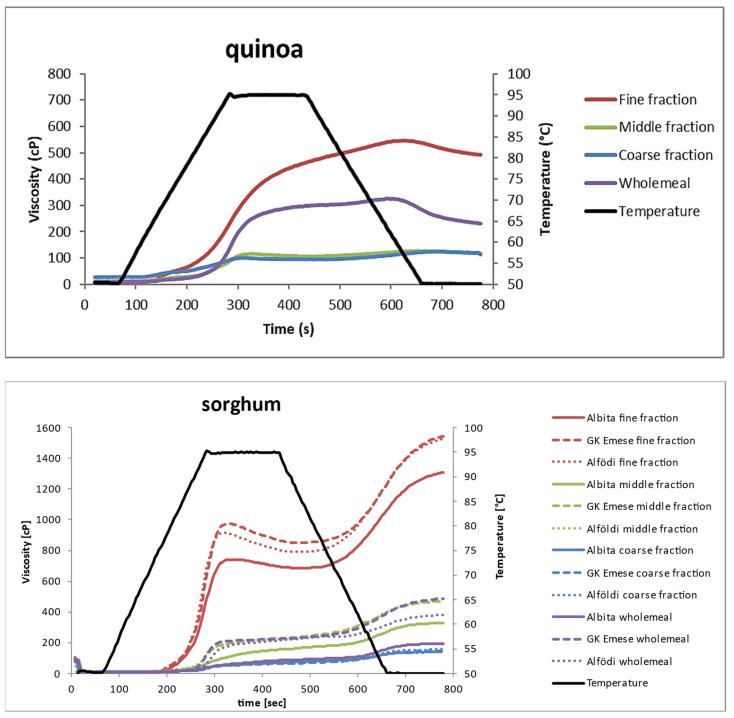
Pasting properties (determined by RVA) of the quinoa and sorghum milling fractions.

**Figure 7 foods-12-03125-f007:**
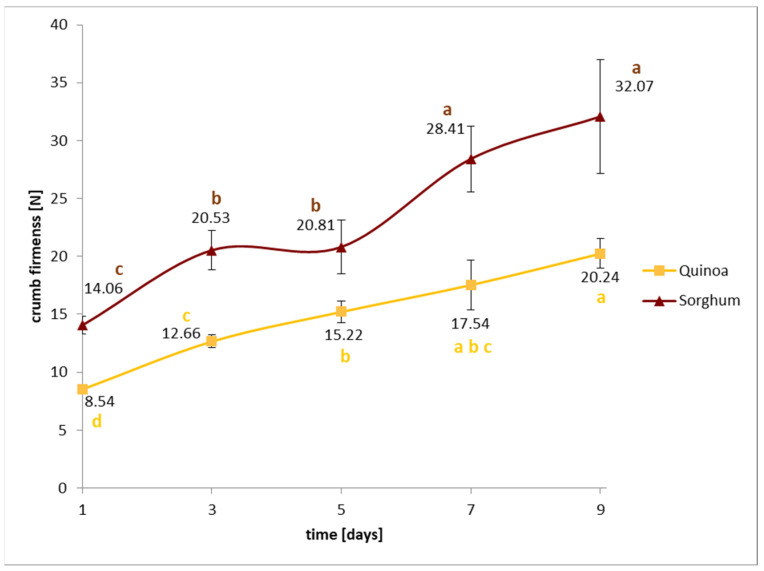
Storage tests for quinoa and sorghum sourdough breads. Different letters above or below each value visualise significant differences between the values within same raw material.

**Figure 8 foods-12-03125-f008:**
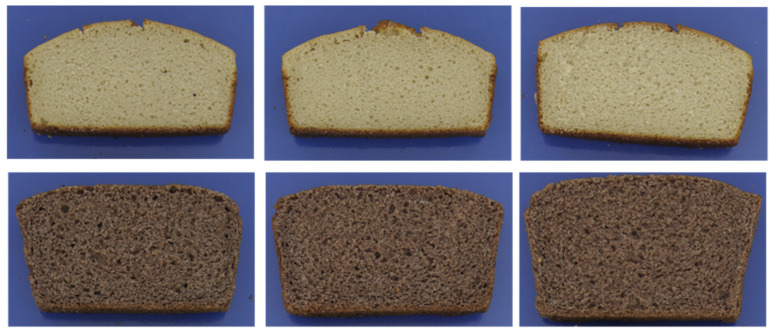
Gluten-free breads from quinoa (**top**) and sorghum (**bottom**), with 70–80–90% dough moisture from left to right.

**Table 1 foods-12-03125-t001:** Ash content of the quinoa and sorghum fractions obtained in the pre-trials (mg/100 g dm, *n* = 3); different superscript letters indicate significant differences between the milling fractions (within same gap setting).

Gap	Coarse Fraction (>475 µm)	Medium Fraction (212–475 µm)	Fine Fraction (<212 µm)
Quinoa			
0	3.1 ± 0.07 ^c^	1.6 ± 0.01 ^b^	0.8 ± 0.04 ^a^
5	4.6 ± 0.55 ^c^	2.2 ± 0.15 ^b^	0.8 ± 0.04 ^a^
8	4.5 ± 0.27 ^c^	2.9 ± 0.00 ^b^	0.9 ± 0.06 ^a^
10	3.6 ± 0.26 ^c^	3.1 ± 0.03 ^bc^	1.4 ± 0.0 ^a^
Sorghum (Albita)			
3	2.46 ± 0.206 ^b^	1.07 ± 0.085 ^a^	1.38 ± 0.048 ^a^
5	2.45 ± 0.189 ^c^	1.23 ± 0.107 ^a^	1.58 ± 0.065 ^b^
7	1.99 ± 0.038 ^a^	1.27 ± 0.014 ^b^	1.89 ± 0.12 ^a^

**Table 2 foods-12-03125-t002:** Physical properties of quinoa and sorghum breads. Results are average values of at least triplicate values ± standard deviation; superscript lowercase letters indicate significant differences between the different dough moistures.

		Water Addition	Quinoa	Sorghum
Baking loss [%, *n* = 4]		70	12.57± 0.81 ^a^	13.76 ± 0.97 ^a^
		80	13.19 ± 0.37 ^ab^	14.86 ±0.29 ^ab^
		90	14.20 ± 0.39 ^b^	15.40 ± 0.29 ^b^
Specific volume [cm³/g, *n* = 4]		70	1.78 ± 0.03 ^a^	2.20 ± 0.05 ^a^
		80	1.78 ± 0.06 ^a^	2.23 ± 0.02 ^a^
		90	1.86 ± 0.03 ^b^	2.46 ± 0.02 ^b^
Texture	Crumb firmness F_max_ [N, *n* = 6]	70	17.16 ± 0.77 ^c^	17.04 ± 1.32 ^b^
		80	13.08 ± 0.22 ^b^	15.39 ± 1.07 ^b^
		90	10.20 ± 0.03 ^a^	10.23 ± 0.02 ^a^
	Relative elasticity [%, *n* = 6]	70	59.88 ± 0.54 ^a^	51.27 ± 1.53 ^a^
		80	62.29 ± 0.38 ^b^	47.58 ± 2.00 ^a^
		90	64.99 ± 0.48 ^c^	53.99 ± 0.22 ^b^
Colour crust	L* [*n* = 4]	70	47.82 ± 1.52 ^a^	37.59 ± 1.10 ^a^
		80	42.84 ± 3.81 ^ab^	35.85 ± 0.96 ^a^
		90	39.28 ± 1.61 ^b^	39.47 ± 1.44 ^b^
	a* [*n* = 4]	70	18.39 ± 0.49 ^a^	13.53 ± 3.41 ^ab^
		80	19.44 ± 0.59 ^a^	13.11 ± 0.16 ^ab^
		90	19.38 ± 0.77 ^a^	14.11 ± 1.70 ^b^
	b* [*n* = 4]	70	35.98 ± 0.32 ^b^	16.12 ± 0.85 ^a^
		80	31.39 ± 2.50 ^ab^	15.89 ± 0.15 ^a^
		90	27.64 ± 1.02 ^a^	18.04 ± 1.96 ^b^
Pore properties	Average pore size [mm², *n* = 8]	70	3.48 ± 0.03 ^a^	11.55 ± 0.63 ^a^
		80	2.76 ± 0.06 ^a^	9.81 ± 1.06 ^a^
		90	3.37 ± 0.39 ^a^	10.94 ± 1.17 ^a^
	Pore area [%, *n* = 8]	70	38.92 ± 2.67 ^a^	48.05 ± 0.78 ^b^
		80	37.50 ± 2.63 ^a^	48.18 ± 0.94 ^b^
		90	37.57 ± 1.61 ^a^	46.41 ± 1.15 ^a^
	Number of pores [*n* = 4]	70	45.17 ± 5.72 ^a^	15.5 ± 1.32 ^a^
		80	51.13 ± 7.68 ^b^	18.38 ± 1.89 ^a^
		90	47.88 ± 3.97 ^ab^	17.38 ± 1.65 ^a^
	Pore uniformity [*n* = 4]	70	3.02 ± 0.58 ^ab^	42.31 ± 3.87 ^ab^
		80	2.32 ± 0.22 ^a^	18.74 ± 3.52 ^a^
		90	3.37 ± 0.18 ^b^	27.33 ± 2.34 ^b^
Colour crumb	L* [*n* = 4]	70	62.14 ± 1.07 ^a^	35.96 ± 0.62 ^b^
		80	64.35 ± 0.68 ^b^	34.47 ± 1.28 ^a^
		90	64.25 ± 1.62 ^b^	37.57 ± 0.76 ^c^
	a* [*n* = 4]	70	6.76 ± 0.14 ^a^	11.06 ± 0.33 ^b^
		80	6.78 ± 0.21 ^a^	10.17 ± 0.30 ^a^
		90	6.71 ± 0.03 ^a^	11.99 ± 0.14 ^c^
	b* [*n* = 4]	70	24.86 ± 0.48 ^a^	12.29 ± 0.17 ^a^
		80	25.23 ± 0.44 ^a^	12.20 ± 0.12 ^a^
		90	25.18 ± 0.57 ^b^	13.05 ± 0.21 ^b^

**Table 3 foods-12-03125-t003:** Avrami parameters for storage tests of quinoa and sorghum breads. (Ɵ = starch ratio, which has not re-crystallised, T_0_ = initial F_max_ at day 0, T_inf_ = final F_max_ at day 9, Tt = Fmax at time “t”, k = rate constant, *n* = Avrami exponent).

Paramter	Quinoa	Sorghum
k	0.049	0.026
n	1.822	2.086
T_inf_ (N)	20.24	32.07
T_0_	8.54	14.06
R²	0.979	0.924

## Data Availability

Data is contained within the article.

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
