# Peer review of "Dry Fractionation and Gluten-Free Sourdough Bread Baking from Quinoa and Sorghum"

_foods, 2023, doi:10.3390/foods12163125_

Round 1

Reviewer 1 Report

General comments: In this paper, effects of milling fractionation on the sourdough fermented bread were studied. The author spent a lot of work including pre-milling trials and chemical composition analysis. The paper was also well written and presented. Other comments: 1. Line 127 “Bread dough was prepared by mixing 100% sourdough with 75% flour (mixture of quinoa or sorghum with gluten -free wheat starch in a ratio of 1:1)”. What does “100% sourdough” mean for the whole bread dough? Were all other ingredients (especially water addition for 70%, 80%, 90% of dough moisture) calculated based on sourdough? Shouldn’t it be total flour (TFW) in bread dough? 2. The bread was stored at 20 oC before physical property testing, but might bread samples have a shelf life of up to 9 days without preservation measures? 3. What about the sensory assessment?

Reviewer 2 Report

The conclusions are weak, especially the first paragraph, which has no scientific contribution.

A control sample is required. It can be wheat flour or other flour for gluten-free products. Alternatively, a commercial sample should have been analyzed for the central assumption that these breads harden more slowly than commercial gluten-free breads.

Line 31-35: Sorghum is a true cereal.

Line 125: 27°C. A space between the degree symbol and the temperature value is required. It is in different places, like line 86, 323, etc.

Line: (starch still made around 50g/100 g dm). According to the international system of units the values are separated from the symbols. Please, check it throughout the document.

Reviewer 3 Report

The manuscript “Dry fractionation and sourdough fermented, gluten-free bread baking from quinoa and sorghum” is focused on assessing the breadmaking quality of different milling fractions of quinoa and sorghum, obtained with a laboratory scale roller mill. In addition, the baking properties and the shelf-life of sourdough bread obtained from 100% quinoa or 100% sorghum were investigated and compared.

The topic of the paper is really interesting and original. The authors make a good argument for the hypothesis question in the Introduction section.

The abstract section is well-designed, and the relevant aspects of the paper are highlighted.

The research methodology was adequately chosen.

The authors discuss relevant bibliographic references for this topic. A critical discussion of the main quality parameters is presented allowing the authors to draw the most important conclusions. The conclusions are consistent with the presented arguments and they address the main question posed.

The manuscript is well-written and logically organized for the topic. The Results and Discussion section is sustained by 2 tables and 8 figures, which are relevant to the topic and well-designed. The manuscript is easy to read and the tables and figures are informative.

Overall, the paper will have a significant contribution to the state of the art in this field. The research results will be a good base for developing quinoa ans sorghum industrial milling and breadmaking methods.

Some minor observations are:

-         Please introduce information about the chemical reagents used in the research

-         Subsection 2.5. Gluten-free bread baking- it is unclear what kind of quinia and sorghum flour (lines 123, 128) was used.

-         Please correct the title of section 3. Results and Discussion (probably this was the authors’ intention) and the numbering of the sections. Thus, Conclusions becomes number 4

-         It is confusing why the authors used two terms: „coarse” and „course” fractions in the paper. I believe the correct term is coarse fraction

-         Lines 357-359- Is there any reason why only sorghum was considered improved after microbial fermentation? The nutritional quality improvement of quinoa food is also well known.

-         Line 408 -the authors stated:  „Quinoa and sorghum were used as wholegrain flours”- is this wholemeal discussed in the sub-section Pasting properties of the milling fractions?

Minor editing of English language required
